# ENO1 Promotes OSCC Migration and Invasion by Orchestrating IL-6 Secretion from Macrophages via a Positive Feedback Loop

**DOI:** 10.3390/ijms24010737

**Published:** 2023-01-01

**Authors:** Ying Lin, Wenwen Zhang, Luyao Liu, Weibo Li, Yafei Li, Bo Li

**Affiliations:** Jilin Provincial Key Laboratory of Oral Biomedical Engineering, Department of Oral Anatomy and Physiology, Hospital of Stomatology, Jilin University, Changchun 130021, China

**Keywords:** oral squamous cell carcinoma, macrophage, alpha-enolase, interleukin-6, lactic acid

## Abstract

Oral squamous cell carcinoma (OSCC) has a five-year survival rate of less than 50% due to its susceptibility to invasion and metastasis. Crosstalk between tumor cells and macrophages has been proven to play a critical role in tumor cell migration and invasion. However, the specific mechanisms by which tumor cells interact with macrophages have not been fully elucidated. This study sought to investigate the regulatory mechanism of tumor cell-derived alpha-enolase (ENO1) in the interaction between tumor cells and macrophages during OSCC progression. Small interfering RNA (siRNA) transfection and recombinant human ENO1 (rhENO1) stimulation were used to interfere with the interaction between tumor cells and macrophages. Our results showed that ENO1 was expressed higher in CAL27 cells than in HaCaT cells and regulated lactic acid release in CAL27 cells. Conditioned medium of macrophages (Macro-CM) significantly up-regulated the ENO1 mRNA expression and protein secretion in CAL27 cells. ENO1 promoted the migration and invasion of tumor cells by facilitating the epithelial–mesenchymal transition (EMT) through macrophages. ENO1 orchestrated the IL-6 secretion of macrophages via tumor cell-derived lactic acid and the paracrine ENO1/Toll-like receptor (TLR4) signaling pathway. In turn, IL-6 promoted the migration and invasion of tumor cells. Collectively, ENO1 promotes tumor cell migration and invasion by orchestrating IL-6 secretion of macrophages via a dual mechanism, thus forming a positive feedback loop to promote OSCC progression. ENO1 might be a promising therapeutic target which is expected to control OSCC progression.

## 1. Introduction

Oral squamous cell carcinoma (OSCC) is a common type of head and neck malignancy with high morbidity and mortality [1,2]. Despite surgical resection combined with chemotherapy and radiation being applied in many cases, the overall five-year survival rate of OSCC patients has not exceeded 50%, and no significant improvement to date, due to its susceptibility to invasion and metastasis [3,4]. Therefore, it is necessary to elucidate the regulatory molecular mechanisms of tumor cell migration and invasion in OSCC. 

Tumor cell migration and invasion are significantly associated with the tumor microenvironment (TME) [5]. Generally, tumor-associated macrophages (TAMs) play prominent roles in the courses of tumorigenesis and malignant transformation by secreting cytokines and chemokines [6,7]. The density of TAMs correlates positively with histological grade and negatively with prognosis [8,9,10]. Macrophages and tumor cells exert reciprocal influence to regulate tumor progression. Thus, some secretory proteins can not only promote the migration and invasion of tumor cells via autocrine signaling pathways, but they can also induce macrophage activation and orchestrate the secretion of inflammatory cytokines via paracrine signaling pathways, which can in turn post positive feedback with tumor cells, thus promoting tumor cell migration and invasion.

Alpha-enolase (ENO1), a rate-limiting glycolytic enzyme, is considered as a multifunctional protein that participates in various intracellular and extracellular activities, depending on its subcellular localization [11]. Firstly, the primary function of ENO1 is to catalyze glycolysis, which promotes lactic acid release in the Warburg effect [11,12]. Secondly, in the cytoplasm, ENO1 maintains mitochondrial membrane stability. Moreover, it can also regulate multiple intracellular signaling pathways such as HIF-1α, PI3K/AKT, Wnt/β-catenin pathway and so on [13,14,15]. In the nucleus, ENO1 serves as an RNA binding protein by interacting with circ-RNA [16]. Thirdly, ENO1 acts as a plasminogen receptor and promotes extracellular matrix (ECM) degradation when it is localized on the cell membrane [17]. Notably, in the extracellular environment, ENO1 is associated with exosomes or secreted as a soluble protein [18,19,20]. Our study focuses on the role of ENO1 together with lactic acid in the interaction between tumor cells and macrophages. However, how extracellular ENO1 could activate macrophages remains unclear. In the present study, small interfering RNA (siRNA) transfection and recombinant human ENO1 (rhENO1) stimulation were used to investigate the role and mechanism of tumor cell-derived ENO1 in the interaction between tumor cells and macrophages during OSCC progression.

## 2. Results

### 2.1. Expression and Secretion of ENO1 in Tumor Cells and Its Regulation of Lactic Acid Release

To evaluate the tumor-promoting function of ENO1, the expression of ENO1 was compared between a human tongue squamous cell carcinoma cell line CAL27 and an immortalized normal epithelial cell line HaCaT. The mRNA expression levels of ENO1 were significantly higher in CAL27 cells, compared with HaCaT cells (Figure 1A). To obtain tumor-conditioned macrophages, RAW264.7 were cultured with tumor-conditioned medium (TCM) of CAL27 cells for 24 h. Conditioned medium from tumor-conditioned macrophages (Macro-CM) was collected to incubate OSCC cells. As shown in the results, Macro-CM significantly up-regulated the mRNA expression of ENO1 in CAL27 cells (Figure 1A). According to the Western blot analysis, the protein expression levels of ENO1 were significantly higher in CAL27 cells than in HaCaT cells, although the protein expression levels of ENO1 in CAL27 cells with Macro-CM stimulation showed no significant change (Figure 1B,C). Furthermore, ELISA confirmed that CAL27 cells with Macro-CM stimulation secreted higher levels of ENO1 than that cultured with normal medium, whereas it was not detectable in HaCaT cells (Figure 1D). To further verify the regulatory role of ENO1 on lactic acid release in tumor cells, CAL27 cells were transfected with the ENO1 siRNA (si-ENO1). The scrambled siRNA served as the negative control. The transfection efficiency was confirmed by RT-qPCR (Figure 1E) and Western blot (Figure 1F,G). Significantly, knockdown of ENO1 significantly reduced the release of lactic acid in CAL27 cells (Figure 1H). These findings demonstrated the elevated expression and secretion of ENO1 in tumor cells and its regulation of lactic acid release. 

### 2.2. ENO1 Promotes Tumor Cell Migration and Invasion through Macrophages

The effects of ENO1 on the migration and invasion of tumor cells were determined by wound-healing assay and transwell assay in vitro. To verify the tumor-promoting role of ENO1-mediated crosstalk between macrophages and tumor cells in OSCC, CAL27 cells were transfected with the ENO1 siRNA. Then, RAW264.7 cells were stimulated with tumor-conditioned medium (TCM) of transfected CAL27 cells. Conditioned medium (CM) from macrophages was collected and used to incubate CAL27 cells. The wound-healing assay showed that CM from macrophages induced by ENO1-siRNA-treated CAL27 cells significantly decreased the migration capacity of CAL27 cells (Figure 2A,C). Significantly, knockdown of ENO1 reduced the migration and invasion property of CAL27 cells through macrophages (Figure 2B,D,E). Furthermore, to mimic the TME with ENO1 overexpression, TCM supplemented with rhENO1 was used to induce macrophages. As is shown in the results (Figure 2F,H), the wound-healing assay showed that CM from rhENO1-induced macrophages significantly increased the migration capacity of tumor cells. Accordingly, the transwell assay for migration showed consistent results (Figure 2G,I). Moreover, the transwell assay with matrigel demonstrated that more invasive cells were detected in the rhENO1-treated group than the control group (Figure 2G,J). These results illustrated the importance of ENO1 as a driver of tumor cell migration and invasion in OSCC with the involvement of macrophages. 

### 2.3. ENO1 Promotes Epithelial–Mesenchymal Transition of Tumor Cells through Macrophages

Epithelial–mesenchymal transformation (EMT) is a biological process in which epithelial cells are transformed into cells with mesenchymal phenotypes through a specific procedure [21]. Recent studies have found that ENO1 participates in the EMT-regulating process in lung cancer, gastric cancer, and breast cancer [22]. Therefore, we sought to determine whether ENO1 could promote EMT through macrophages. Macrophages were stimulated by tumor-conditioned medium (TCM) of transfected CAL27 cells. Then, Macro-CM was collected and used to incubate CAL27 cells. Western blot was applied to detect the protein expression of the epithelial marker E-cadherin and the mesenchymal markers N-cadherin and Vimentin in CAL27 cells. The expression of E-cadherin increased and Vimentin decreased in the ENO1-siRNA-treated group, while N-cadherin did not change significantly (Figure 3A,C–E). However, the CM of rhENO1-induced macrophages promoted the expression of mesenchymal markers N-cadherin and Vimentin in CAL27 cells, whereas it reduced the epithelial marker E-cadherin expression of CAL27 cells (Figure 3B,F–H). Overall, these results indicated that ENO1 is an important mediator for promoting the EMT of tumor cells through macrophages.

### 2.4. ENO1 Orchestrates IL-6 Secretion of Macrophages 

To elucidate the factors involved in macrophage activation in OSCC, we measured the gene expression of five cytokines including IL-6, Il-10, IL-12β, TNF-α and TGF-β using RT-qPCR. We found that the mRNA levels of IL-6 were significantly decreased and IL-10 was increased in macrophages induced by the TCM of ENO1-siRNA-transfected CAL27 cells for 6 h and 12 h, respectively (Figure 4A,B). Consistently, IL-6 secretion levels were down-regulated in macrophages induced by the TCM of ENO1 siRNA-treated CAL27 cells for 24 h (Figure 4E). On the contrary, the mRNA levels of IL-6 were significantly increased in rhENO1-induced macrophages for 6 h and 12 h, respectively (Figure 4C,D). Moreover, IL-6 protein levels were up-regulated in rhENO1-induced macrophages for 24 h, which was confirmed by ELISA (Figure 4F). However, the mRNA levels of IL-10 were markedly decreased in rhENO1-induced macrophages for 6 h (Figure 4C). Fold changes of other cytokines showed no statistical differences. These results indicated that IL-6 was a potential downstream target of ENO1 in macrophages. 

### 2.5. ENO1 Orchestrates IL-6 Secretion of Macrophages via Tumor Cell-Derived Lactic Acid

ENO1 is a key regulator participating in the activation of macrophages and the promotion of tumor progression. ENO1 can regulate the release of lactic acid, which is a vital regulator of macrophage activation [23]. Therefore, we further investigated the regulatory effect of lactic acid on the cytokines of macrophages. Our study focused on IL-6, which is abundant in the OSCC microenvironment. TCM, supplemented with exogenous lactic acid or monocarboxylate transporters (MCTs) inhibitor, was applied to stimulate macrophages. α-cyano-4 hydroxycinnamate (α-CHC) is a classical inhibitor of MCTs, which is 10 times more selective for MCT-1 than for other MCTs, preventing the uptake of lactic acid into cells. First, the CCK8 assay was performed to detect the cytotoxicity of lactic acid and α-CHC, and safe concentrations were selected for further experiments (Figure 5A,B). We found that 10 mM lactic acid and 1 mM α-CHC were relatively safe concentrations for RAW264.7 cells. Exogenous lactic acid significantly increased IL-6 mRNA levels (Figure 5C), while 1 mM α-CHC inhibited IL-6 mRNA levels compared with the solvent control group (Figure 5D). Consistently, as determined by ELISA, the concentration of IL-6 was elevated significantly in response to treatment with 10 mM lactic acid compared with the control group (Figure 5E), while α-CHC inhibited IL-6 secretion levels compared with the solvent control group (Figure 5F). Collectively, ENO1 orchestrated the IL-6 secretion of macrophages via tumor cell-derived lactic acid.

### 2.6. ENO1 Orchestrates IL-6 Secretion of Macrophages via Paracrine ENO1/TLR4 Signaling 

Macrophages can mediate the ENO1-induced migration and invasion of CAL27 cells, while the underlying mechanism has not been elucidated. Interestingly, paracrine ENO1 tends to activate the CD14-dependent TLR4 pathway via functionally binding with TLR4 on monocytes in rheumatoid arthritis (RA) [20]. However, there is limited evidence supporting the link between tumor-derived paracrine ENO1 and TLR4 expression in macrophages. It has been reported that TLR4 on the macrophage surface plays an important role in the polarization of macrophages. To explore whether TLR4 in macrophages mediates the paracrine ENO1-mediated macrophage activation and secretion of IL-6, TLR4 inhibitor (TAK-242) was applied to evaluate the effect of exogenous ENO1 on IL-6 expression at the mRNA and protein levels. Results showed that TLR4 inhibitor markedly suppressed the mRNA and protein expression of IL-6 in rhENO1-stimulated macrophages (Figure 6A,B). Furthermore, immunofluorescence staining showed the colocalization of ENO1 and TLR4 on macrophages after stimulation of TCM supplemented with rhENO1 (Figure 6C). Our results suggested that ENO1 orchestrated IL-6 secretion of macrophages via paracrine ENO1/TLR4 signaling.

### 2.7. IL-6 Promotes the Migration and Invasion of Tumor Cells 

The IL-6 receptor (IL-6R) antagonist tocilizumab (Toc) was applied to verify the role of IL-6/IL-6R signaling during tumor cell migration and invasion. CAL27 cells were pretreated with IL-6R antagonist tocilizumab to inhibit IL-6R. RAW264.7 cells were incubated with TCM for 24 h to generate tumor-conditioned macrophages in advance. Then, CAL27 cells were cocultured with macrophages in the transwell chamber. Transwell assays showed that the tocilizumab treatment significantly suppressed the migration and invasion properties of tumor cells compared with the control group (Figure 7A,B). These results indicated that IL-6/IL-6R signaling could enhance the tumor cell migration and invasion of tumor cells in OSCC. 

## 3. Discussion

In the present study, we demonstrated that ENO1 was highly expressed in OSCC cells and was crucial for maintaining pro-tumoral properties via orchestrating the secretion of IL-6 in macrophages. Mechanically, ENO1 can not only orchestrate IL-6 expression via lactic acid, but also activate macrophages via the ENO1/TLR4 paracrine signaling pathway.

Although the direct functions of ENO1 in tumor cells have been extensively reported [24], paracrine ENO1 plays a critical role in the crosstalk between tumor cells and macrophages, which has not been clearly elucidated. The previous study revealed that ENO1 was expressed at an elevated level in OSCC tissues, compared with corresponding normal counterpart [19]. Overexpression of ENO1 was proved to be correlated with poor prognosis in various types of cancers including breast cancer [25], gastric cancer [26], bladder cancer [27], glioma [14] and non-small cell lung cancer [28]. Notably, ENO1 plays a pro-tumoral role in OSCC through the circ-AMOTL1/miR-22-3p/miR-1294 network [16]. In addition, exosome-derived ENO1 regulates integrin α6β4 expression and accelerates hepatocellular carcinoma (HCC) growth and metastasis [18,19]. Hence, the inhibition of ENO1 by a unique small molecule inhibitor AP-III-a4 (ENOblock) could reduce the migration and invasion abilities of tumor cells in gastric cancer [26]. These findings suggest that the highly expressed ENO1 in tumor cells may take part in the interaction between tumor cells and macrophages, which accounts for the tumor progression.

In addition to differentially expressed cytokines or secretory proteins in TME, the tumor-promoting capability of macrophages was activated by lactic acid, which is known to promote M2-like polarization [23,29]. Lactic acid can facilitate tumor progression through paracrine signaling mechanisms involving suppression of immune surveillance [30,31]. The mechanism of lactic acid induction of macrophages involves multiple pathways, including HIF-1α [32], PI3K-AKT [33], NF-κB [34], STAT-3 [35], STAT-6 [36] and histone lactylation [29]. Lactic acid elevated vascular endothelial growth factor (VEGF) expression of TAMs [23]. Lactic acid-activated macrophages promoted the invasive property of pituitary adenoma via CCL17 [36]. As prior studies indicated, lactic acid produced by PKM2 up-regulation promoted Galectin-9-mediated immunosuppression via NF-κB signaling inhibition in HNSCC [37]. Our results are consistent with previous studies showing that lactic acid in the tumor-conditioned medium can orchestrate pro-tumoral cytokine production, such as IL-6 [38]. 

TLR4, recognized as a pathogen-associated molecular pattern (PAMP), expresses on immune cells and tumor cells. Nevertheless, TLR4 activation generally plays an ambivalent role during tumor progression [39]. On one hand, TLR4 is correlated with IL-6 expression and poor prognosis in primary breast carcinoma, but on the other hand, TLR4 stimulation reduces microglia-assisted breast cancer cell invasion and TLR4 stimulation alters the macrophage or microglia response in brain metastasis of breast carcinoma [39]. Despite the fact that there are a great number of TLR4 activators in the TME, there is limited evidence supporting the link between tumor-derived paracrine ENO1 and TLR4 expression. Interestingly, paracrine ENO1 tends to activate the CD14-dependent TLR4 pathway via functionally binding with TLR4 on monocytes by a dual mechanism, initially pro-inflammatory and later anti-inflammatory, in rheumatoid arthritis (RA) [20]. Heat-shock proteins (HSPs) are identified as damage-associated molecular patterns (DAMPs) to functionally bind to TLR4 signaling pathways in diverse settings. For example, HSP70 protein was found to activate the TLR4/NF-kB pathway in macrophages. Similarly, extracellular HSP90α can activate MyD88-IRAK-complex-associated NF-κB and STAT-3 signaling in macrophages for pro-tumoral M2-like polarization [5]. ENO1 may be one of the paracrine biomolecules in the packages of the extracellular vesicles (EVs) or exosomes [18,19]. However, whether ENO1 has a similar role to HSPs when undergoing acidified and hypoxic stress in TME, the underlying mechanism and relevant pathway remain to be studied more specifically. 

According to our research, we found that IL-6 was significantly elevated in macrophages induced by ENO1-treated macrophage-conditioned medium or exogenous lactic acid. Intriguing studies implicated that TAM-derived IL-6 supports tumor progression in several settings [5,40,41,42,43]. IL-6 is an important interleukin of chronic inflammation that binds to IL-6R, which results in the activation of the transcription factor STAT3. IL-6/STAT3 signaling promotes tumor cell proliferation, metastasis, angiogenesis, immune suppression, cancer stemness and chemotherapeutic resistance [44,45,46]. IL-6 may promote cancer cell migration and invasion by enhancing the PLOD2-integrin β1 signaling pathway in OSCC cells [47]. The efficacy of STAT3-associated inhibitors has been verified in vitro and in vivo, and molecules involved in the STAT3 pathway are expected to be a promising target for the treatment of OSCC [48]. Our observations revealed that the silence of the ENO1 gene decreased the release of lactic acid and inhibited macrophages to secrete IL-6 in vitro. Therefore, these results indicated that ENO1-mediated lactic acid release could up-regulate IL-6 expression of macrophages, which promoted migration, invasion and EMT of tumor cells in turn.

However, the results showed that ENO1 inhibited IL-10 mRNA levels in the early stage of tumor-conditioned medium (TCM) incubation. The change in IL-10 levels could be explained as follows. Firstly, TCM-activated macrophages are a mixed group of M1 and M2-like subsets [49]. There are some factors to promote M1 or M2-like phenotype in TCM. The cytokine expression profiles of TAMs are spatially and temporally diverse. Secondly, ENO1 promotes M1-like polarization in the early stage. The overall effect is to promote tumor progression, as M1-like macrophages could cascade a stem-like phenotype of tumor cells via the IL6/Stat3/THBS1 feedback loop [50]. It seems not contradictory to ENO1. Thirdly, even though lactic acid promotes IL-10 expression, there may be a time gap between lactic acid production and the effect of rhENO1 on IL-10 expression in macrophages under the incubation of TCM. Fourthly, the expression of different cytokines has a time difference. In the model of rheumatoid arthritis (RA), ENO1 induces early production of pro-inflammatory cytokines and chemokines but delays production of IL-10 to activate the innate immune system [20]. The mechanism of ENO1 regulating IL-10 has not been revealed yet, which will be the content of our subsequent study.

In the TME, the interaction between tumor cells and macrophages is the highlight of the current study. Reprogramming TAMs from a protumor phenotype towards an antitumor phenotype is a promising avenue for the treatment of OSCC [49]. As Taniguchi, S. et al. reported, tumor stem cells set up an IL-33/TGF-β signaling loop to accelerate tumor progression [51]. In turn, IL-33-responding macrophages transmit paracrine TGF-β feedback signals to tumor cells, facilitating invasiveness and drug resistance and further up-regulating IL-33 expression [51]. Moreover, Hou et al. displayed a positive feedback loop in which HCC cell-derived paracrine PKM2 induced macrophage differentiation, and then macrophage-derived CCL1 further enhanced the secretion of PKM2 in a CCL1/CCR8 axis-dependent manner, thus promoting tumor genesis and development [52]. Our study showed that the synergistic effect of paracrine ENO1 and lactic acid promoted the secretion of some pro-tumoral cytokines, such as IL-6, which was confirmed to be highly expressed in macrophages in OSCC. In general, IL-6 promoted tumor cell migration and invasion by enhancing the EMT properties of tumor cells. IL-6 could also induce stem cell properties and even further enhance aerobic glycolysis of tumor cells in turn [40]. 

In conclusion, ENO1 promotes tumor cell migration and invasion by orchestrating macrophage-derived IL-6 via secretion of lactic acid and extracellular ENO1, thus forming a positive feedback loop to promote OSCC progression. ENO1 might be a promising therapeutic target which is expected to control OSCC progression.

## 4. Materials and Methods

### 4.1. Cell Culture

The human oral squamous cell carcinoma cell line CAL27, the human immortalized keratinocyte cell line HaCaT and the mouse macrophage cell line RAW264.7 cells were obtained from the China Center for Type Culture Collection (CCTCC, Wuhan, China). CAL27, HaCaT and RAW264.7 cells were maintained in Dulbecco’s modified Eagle’s medium (DMEM, BI, Israel) supplemented with 10% fetal bovine serum (FBS, BI, Israel), 1% penicillin and streptomycin (BI, Israel) at 37 °C in a humidified atmosphere with 5% CO_2_. In the indicated experiments, CAL27 cells were cultured in a medium with 5 μg/mL IL-6 receptor inhibitor Tocilizumab (Toc, A2012, Selleck, Houston, TX, USA).

### 4.2. Activation of Macrophages and Preparation of Conditioned Medium (CM)

Tumor-conditioned medium (TCM) was prepared as follows. Tumor cells grown to 80% confluence were incubated with DMEM for 24 h; the supernatant was collected, centrifuged at 1000 rpm for 10 min, passed through 0.2 mm filters and stored at −80 °C for future experiments. To obtain tumor-conditioned macrophages, RAW264.7 were cultured with 70% TCM from CAL27 cells for 24 h. Then, the TCM was replaced with complete medium, and macrophage-conditioned medium (Macro-CM) was collected after additional 24 h, centrifuged 1000 rpm for 10 min, passed through 0.22 mm filters, and stored at −80 °C for future experiments.

### 4.3. In Vitro Stimulation of Macrophages

RAW264.7 were plated at 2.5 × 10^5^ cells per well in a sterile six-well culture plate in 2 mL DMEM. The macrophage stimulation experiments included the following conditions: TCM (control), 10 mM lactic acid (L6402, Sigma-Aldrich, St. Louis, MO, USA) in TCM; 1% DMSO (solvent control) and 1 mM α-CHC (S8612, Selleck, Houston, TX, USA) in TCM; 100 ng/mL rhENO1 (ab89248, Abcam, Cambridge, MA, USA) in TCM; 1 μM TAK242 (614316, Sigma-Aldrich, St. Louis, MO, USA) together with 100 ng/mL rhENO1 in TCM.

### 4.4. Cell Transfection

CAL27 cells were seeded into six-well plates (1 × 10^5^ cells/well) and grown for 24 h. Small interfering RNA (siRNA)-targeting ENO1 (si-ENO1) and negative controls (scrambled siRNA) were transiently transfected into OSCC cell lines using GP-transfect-mate (Gene pharma, Suzhou, China) according to the manufacturer’s instructions. The targeting siRNA sequence is listed on Appendix A. Cells cultured with complete medium were used as untreated control. The medium was replaced with serum-free DMEM after 24 h transfection. The transfected cells and supernatant were collected after another 48 h for further experiments. The transfection efficiency was detected using RT-qPCR in 48 h and Western blot in 72 h.

### 4.5. Determination of Lactic Acid Concentration

The lactic acid concentration was measured using a Lactate Assay Kit (K627, BioVision, Milpitas, CA, USA) according to the manufacturer’s instructions. Conditioned media were prepared as triple replicates for the colorimetric lactate assay. The absorbance was measured at 450 nm immediately after incubating the reaction for 30 min at room temperature. Background absorbance was subtracted. The mean values for lactate concentration (mM) were calculated for each condition.

### 4.6. Cell Viability Assay

A total of 2 × 10^3^ cells/well were seeded in 96-well plates and cultured for 24 h. The cells were then treated with a series of gradient concentrations of lactic acid or α-CHC in the presence of TCM. At the indicated time points, cell viability was measured using a cell counting kit-8 (CCK-8, APEXBIO, Boston, MA, USA) assay in accordance with the manufacturer’s instructions. The absorbance values were measured at 450 nm.

### 4.7. Wound-Healing Assay

After overnight starvation with serum-free DMEM medium, straight-line scratches were made to CAL27 cells by a 200 µL micropipette tip, and the cells were then washed to remove detached cells and debris. At 0 and 24 h, photographs of the same area were taken, and the closure of the wound was measured after incubation of the serum-free CM from different conditional macrophages. ImageJ software was used to analyze the wound’s area. The healing rate was calculated as (%) = (initial average scratch area-average scratch area at 24 h)/initial average scratch area × 100%.

### 4.8. Transwell Assay

The assay was conducted using transwell inserts (8 μm pore size, Corning, 3422, USA) in 24-well dishes. The supernatants from transfected CAL27 cells (scrambled siRNA, ENO1 siRNA) cultured in serum-free DMEM were harvested after 48 h. Then, RAW264.7 cells were seeded into the lower chamber of a 24-well plate and then induced with the above supernatants for 24 h. CAL27 cells (5 × 10^4^ per well) were added to the upper compartment of a 24-well transwell plate and cocultured for 24 h with FBS-free DMEM. The lower compartment contained DMEM with 15% FBS. The migrated cells were fixed, stained with crystal violet (Sigma-Aldrich, St Louis, MO, USA) and photographed under a light microscope.

CAL27 cells (5 × 10^4^ per well) were added to the upper compartment of a 24-well transwell chamber coated with Matrigel and cocultured for 48 h with FBS-free DMEM. The lower compartment contained DMEM with 15% FBS. Macrophages were activated by TCM in the lower compartment in advance. The invasive cells were fixed, stained with crystal violet (Sigma-Aldrich, St Louis, MO, USA) and photographed under a light microscope (Nikon, Tokyo, Japan).

### 4.9. RNA Extraction and Real-Time Quantitative PCR (RT-qPCR)

Total RNA was isolated using Trizol reagent (Takara, Tokyo, Japan) and cDNA was synthesized using the RR047A kit (Takara, Tokyo, Japan) according to the manufacturer’s instructions. RT-qPCR was carried out in an Applied Biosystems QuantStudio 3 Real-Time PCR System (Waltham, MA, USA) with the RR820A kit (Takara, Tokyo, Japan). The primers were designed and synthesized in Sangon Biotech (Shanghai, China). The primers are listed in Appendix A. β-actin was used as the endogenous control, and the 2^−ΔΔCT^ method was used to analyze the relative gene expression data.

### 4.10. Protein Extraction and Western Blot Analysis 

Cell protein was extracted with RIPA lysis buffer supplemented with PMSF and phosphatase inhibitor cocktail A (P1801, Beyotime, Shanghai, China), and total protein was quantified using a BCA Protein Assay Kit (P0012S, Beyotime, Shanghai, China). Equal amounts of protein were loaded onto 10% SDS/PAGE gels and transferred to PVDF membrane (Millipore, Billerica, MA, USA). Then, the membranes were blocked with 5% skimmed milk in Tris-buffered saline containing 0.1% Tween-20 (TBST), and the PVDF membranes were incubated with the primary antibody overnight at 4℃ followed by secondary antibody incubation. The primary antibodies used in this study include ENO1 (1: 5000; ab155102, Abcam, Cambridge, MA, USA); anti-E-cadherin (1:10,000; ab40772, Abcam, Cambridge, MA, USA), anti-N-cadherin (1:5000; ab76011, Abcam, Cambridge, MA, USA), anti-Vimentin (1:2000; ab92547, Abcam, Cambridge, MA, USA), and anti-β-actin (ab115777, 1:2000, Abcam, Cambridge, MA, USA). The secondary antibody Dylight 800 Goat Anti-Rabbit IgG (1:5000, A23920, Abbkine, Wuhan, China) was used for 1 h incubation at room temperature. Images were acquired by Odyssey CLX (LICOR, Lincoln, NE, USA), and band analyses were performed using IMAGE J software (NIH).

### 4.11. Enzyme-Linked Immunosorbent Assay (ELISA)

The levels of human ENO1 and murine IL-6 in the cell culture supernatants were measured by ELISA kits (Cusabio, Wuhan, China), following the manufacturer’s instructions. All samples were conducted in duplicate.

### 4.12. Immunofluorescence (IF)

For the immunofluorescent staining experiments, the RAW264.7 cells were incubated with 1 μg/mL rhENO1 in TCM for 30 min. The cells were fixed in 4% PFA for 15 min and washed three times with PBS. Samples were blocked with 5% albumin from bovine serum (BSA) with 0.1% Triton X-100 in PBS for 60 min at room temperature and then incubated with primary antibodies overnight at 4 °C, followed by the secondary fluorescently labeled antibodies for one hour at room temperature. The primary antibodies used in this experiment were: anit-ENO1 (ab155102, 1:500) and anti-TLR4 (ab22048, 1:100) from abcam. The secondary antibodies used in this experiment were: Goat anti-Rabbit IgG (H+L) conjugated with Alexa Fluor 594 (R37117, 2 drops diluted in 1 mL PBS) from Thermo Fisher Scientific, and goat anti-mouse IgG (H+L) conjugated with Alexa Fluor 488 (ab150113, 1:200) from abcam. The nuclei were counterstained with DAPI (Invitrogen, Waltham, MA, USA). Images were captured via an inverted fluorescence microscope (Nikon, Japan).

### 4.13. Statistical Analysis

All statistical analysis were performed by GraphPad Prism (version 9.0, GraphPad Software, Inc., San Diego, CA, USA). The significant difference between two groups was determined by independent sample Student’s *t*-test. The one-way analysis of variance (ANOVA) was employed to compare more than two groups. A two-sided *p* value < 0.05 was considered significant. The results are expressed as the mean ± SEM from three different independent experiments.

## 5. Conclusions

In conclusion, ENO1 promotes tumor cell migration and invasion by orchestrating macrophage-derived IL-6 via secretion of lactic acid and extracellular ENO1, thus forming a positive feedback loop to promote OSCC progression. ENO1 might be a promising therapeutic target which is expected to control OSCC progression.

## Figures and Tables

**Figure 1 ijms-24-00737-f001:**
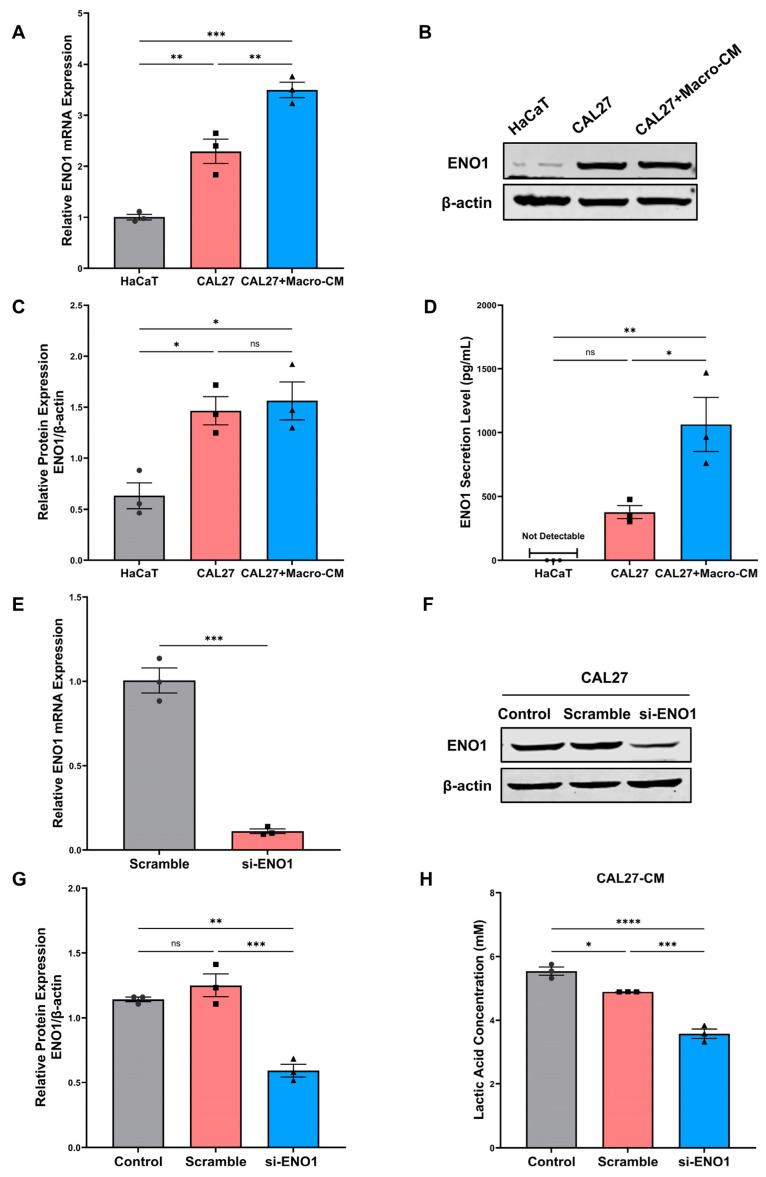
Expression and secretion of ENO1 in tumor cells and its regulation of lactic acid release. (**A**) Relative mRNA expression levels of ENO1 were detected by RT-qPCR in different cell lines. β-actin was used as a reference to normalize the data. (**B**,**C**) Relative protein expression levels of ENO1 were detected by Western blot. β-actin was used as a reference to normalize the data. (**D**) Protein secretion levels of ENO1 in cell culture supernatant were detected by ELISA. (**E**) The transfection efficiency of ENO1 mRNA expression levels were detected by RT-qPCR (48 h). (**F**,**G**) The transfection efficiency of ENO1 protein expression levels were detected by Western blot (72 h). (**H**) The lactic acid concentration was determined in the medium of untreated (control), scrambled siRNA and ENO1-siRNA-transfected CAL27 cells. β-actin was used as a reference to normalize the data. Different symbols (circle/square/triangle) were used to represent the data points of independent biological repeated experiments. All data are displayed as mean ± SEM; n = 3; ns, no significance, * *p* < 0.05, ** *p* < 0.01, *** *p* < 0.001 and **** *p* < 0.0001. Different symbols (circle/square/triangle) were used to represent the data points of independent biological repeated experiments.

**Figure 2 ijms-24-00737-f002:**
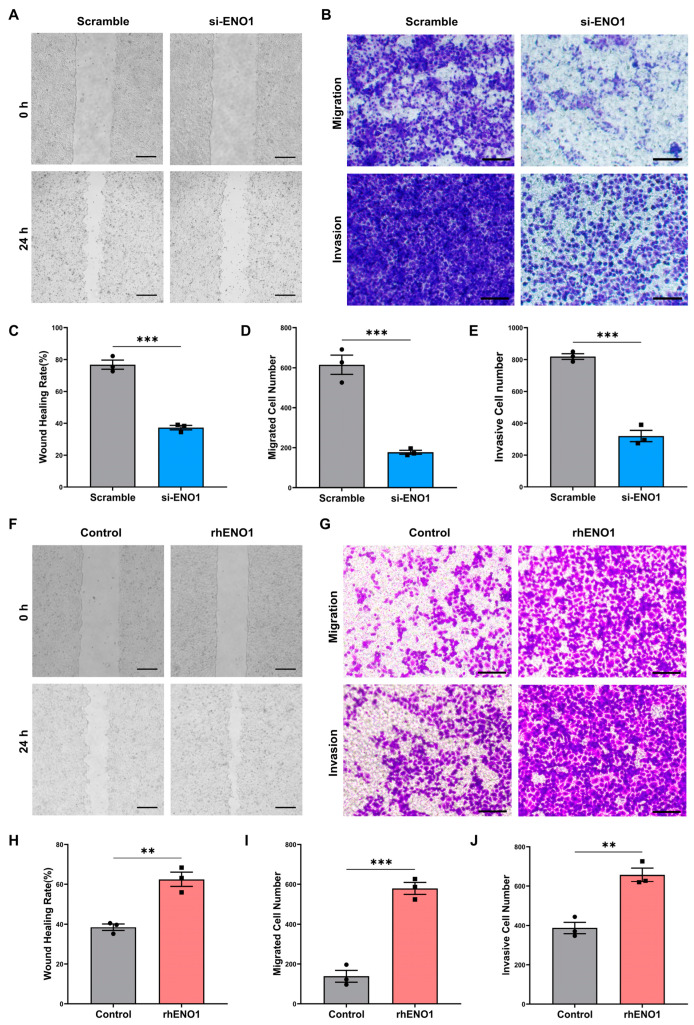
ENO1 promotes tumor cell migration and invasion through macrophages. (**A**,**C**) Wound-healing assay of CAL27 cells incubated with Macro-CM from macrophages induced by TCM of transfected CAL27 cells. (**A**) representative pictures; (**C**) bar charts indicating the wound healing rate. Magnification 40×. Scale bar: 500 μm. (**B**,**D**,**E**) Transwell assay for migration (upper panel) and invasion (lower panel) of CAL27 cells cocultured with macrophages induced by TCM of transfected CAL27 cells. (**B**) representative pictures; (**D**,**E**) bar charts indicating the cell numbers of migrated or invasive cells per area. Magnification 200×. Scale bar: 100 μm. (**F**,**H**) Wound-healing assay of CAL27 cells incubated with Macro-CM from macrophages induced by TCM without or with rhENO1. (**F**) representative pictures; (**H**) bar charts indicating the wound healing rate. Magnification 40×. Scale bar: 500 μm. (**G,I,J**) Transwell assay for migration (upper panel) and invasion (lower panel) of CAL27 cells cocultured with macrophages induced by TCM without or with rhENO1. (**G**) representative pictures; (**I**,**J**) bar charts indicating the cell numbers of migrated or invasive cells per area. Magnification 200×. Scale bar: 100 μm. Different symbols (circle/square) were used to represent the data points of independent biological repeated experiments. All data are displayed as mean ± SEM; n = 3; ** *p* < 0.01 and *** *p* < 0.001.

**Figure 3 ijms-24-00737-f003:**
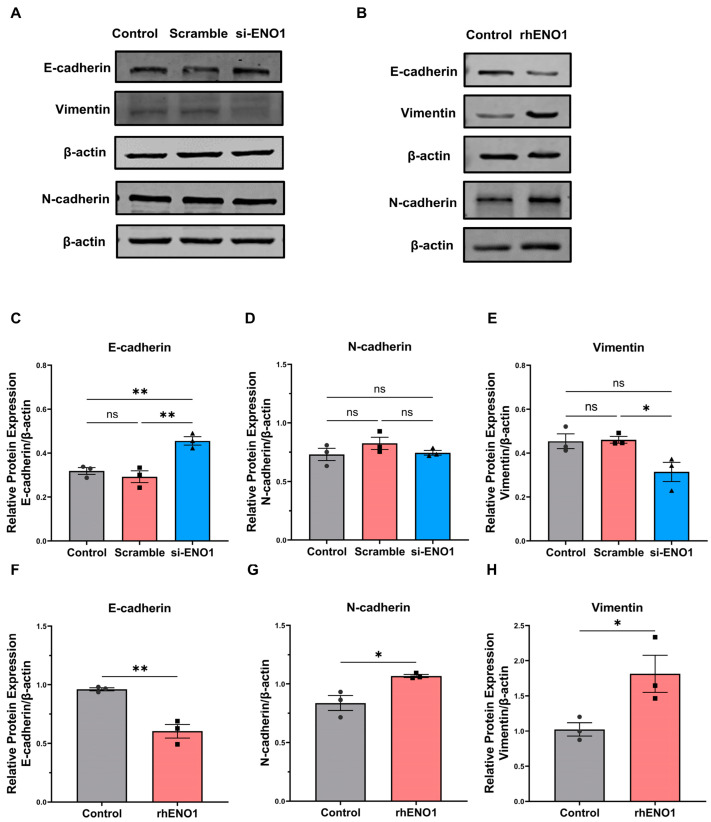
ENO1 promotes epithelial–mesenchymal transition of tumor cells through macrophages. (**A**) Western blot representative image of E-cadherin, Vimentin and N-cadherin relative protein levels in CAL27 cells incubated with Macro-CM from macrophages induced by untreated (control), scrambled siRNA and ENO1-siRNA-transfected tumor supernatant, respectively, for 48 h. (**B**) Western blot representative image of E-cadherin, Vimentin and N-cadherin relative protein levels in CAL27 cells cocultured with Macro-CM from macrophage-induced tumor supernatant without or with rhENO1 for 48 h. (**C**–**H**) Statistical analysis of Western blot. Different symbols (circle/square/triangle) were used to represent the data points of independent biological repeated experiments. All data are displayed as mean ± SEM; n = 3; ns, no significance, * *p* < 0.05 and ** *p* < 0.01.

**Figure 4 ijms-24-00737-f004:**
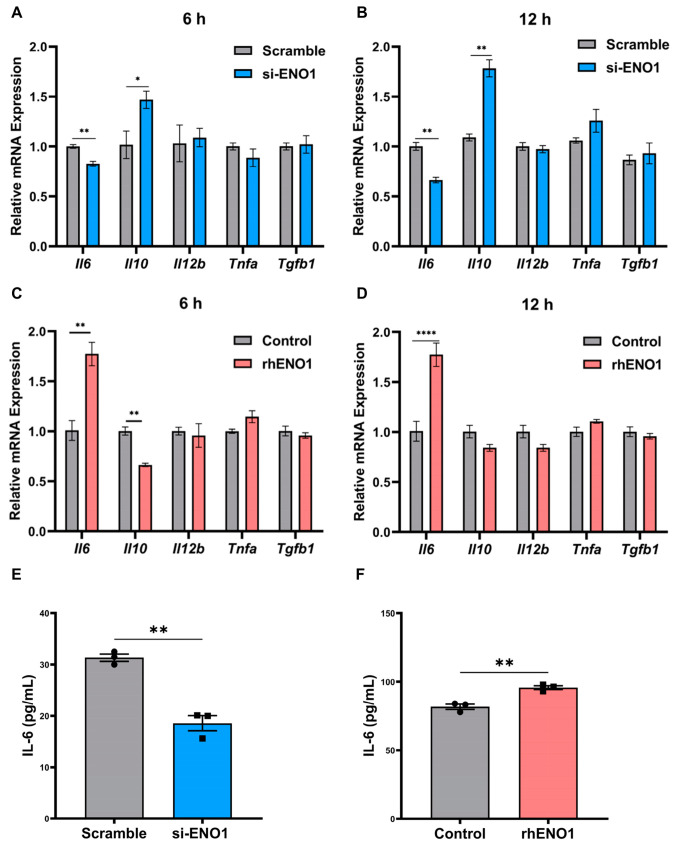
ENO1 orchestrates IL-6 secretion of macrophages. (**A**,**B**) RT-qPCR analysis of the mRNA levels of cytokines (*Il6, Il10, Il12b, Tnfa* and *Tgfb1*) in macrophages incubated with TCM from scrambled siRNA or ENO1-siRNA-transfected CAL27 for 6 h and 12 h, respectively. (**C**,**D**) RT-qPCR analysis of the mRNA levels of cytokines (*Il6, Il10, Il12b, Tnfa* and *Tgfb1*) in macrophages incubated with TCM supplemented without or with rhENO1 for 6 h and 12 h, respectively. (**E**) ELISA for detection of IL-6 protein levels in CM harvested from macrophages incubated with TCM from scrambled siRNA or ENO1-siRNA-transfected CAL27 cells for 24 h. (**F**) ELISA for detection of IL-6 protein levels in CM harvested from macrophages incubated with TCM supplemented without or rhENO1 for 24 h. Different symbols (circle/square) were used to represent the data points of independent biological repeated experiments. All data are displayed as mean ± SEM; n = 3; * *p* < 0.05, ** *p* < 0.01 and **** *p* < 0.0001.

**Figure 5 ijms-24-00737-f005:**
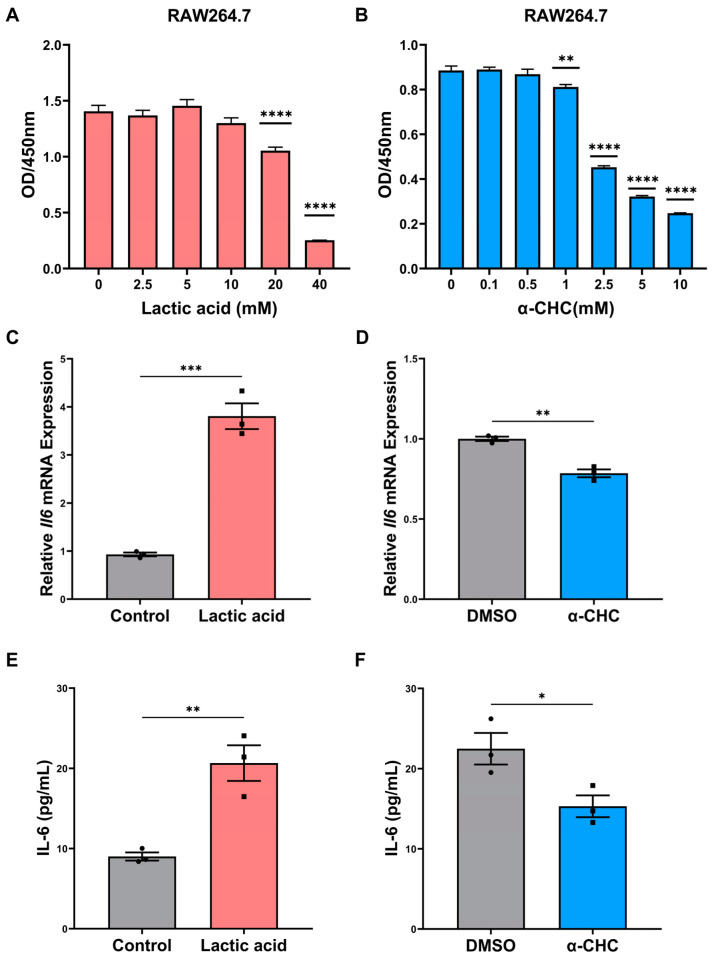
ENO1 orchestrates IL-6 secretion of macrophages via tumor cell-derived lactic acid. (**A**,**B**) CCK8 assay for cell viability of macrophages treated with indicated doses of lactic acid (**A**) or α-CHC (**B**) with TCM for 24 h. (**C**,**D**) RT-qPCR analysis of the mRNA levels of IL-6 in macrophages treated with 10 mM lactic acid (**C**) or 1mM α-CHC (**D**) with TCM for 24 h. (**E**) ELISA for detection of IL-6 protein levels in CM from macrophages treated with TCM without or with 10 mM lactic acid for 24 h. (**F**) ELISA for detection of IL-6 protein levels in CM from macrophages treated with TCM without or with 1 mM α-CHC for 24 h (1%DMSO in TCM was used as a solvent control). Different symbols (circle/square) were used to represent the data points of independent biological repeated experiments. All data are displayed as mean ± SEM; n = 3; * *p* < 0.05, ** *p* < 0.01, *** *p* < 0.001 and **** *p* < 0.0001.

**Figure 6 ijms-24-00737-f006:**
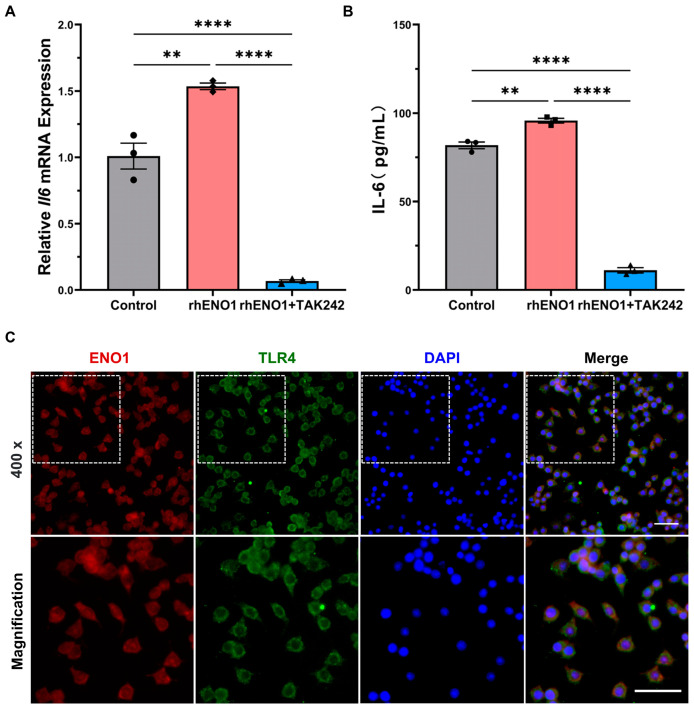
ENO1 orchestrates IL-6 secretion of macrophages via paracrine ENO1/TLR4 signaling pathway. (**A**) Macrophages pretreated without or with TLR4 inhibitor (TAK242, 1 μM) supplemented in TCM were stimulated by rhENO1 for 12 h, and the mRNA levels of IL-6 in macrophages were detected by RT-qPCR analysis. (**B**) Macrophages pretreated without or with TLR4 inhibitor (TAK242, 1 μM) were stimulated by rhENO1 for 24 h, then IL-6 protein levels in CM were assayed by ELISA. (**C**) Immunofluorescence for colocalization of ENO1 and TLR4 on macrophages. Magnification 400×. Scale bar: 50μm. The area indicated on the upper left image was shown in a magnified image on the lower panel. Different symbols (circle/square/triangle) were used to represent the data points of independent biological repeated experiments. All data are displayed as mean ± SEM; n = 3; ** *p* < 0.01 and **** *p* < 0.0001.

**Figure 7 ijms-24-00737-f007:**
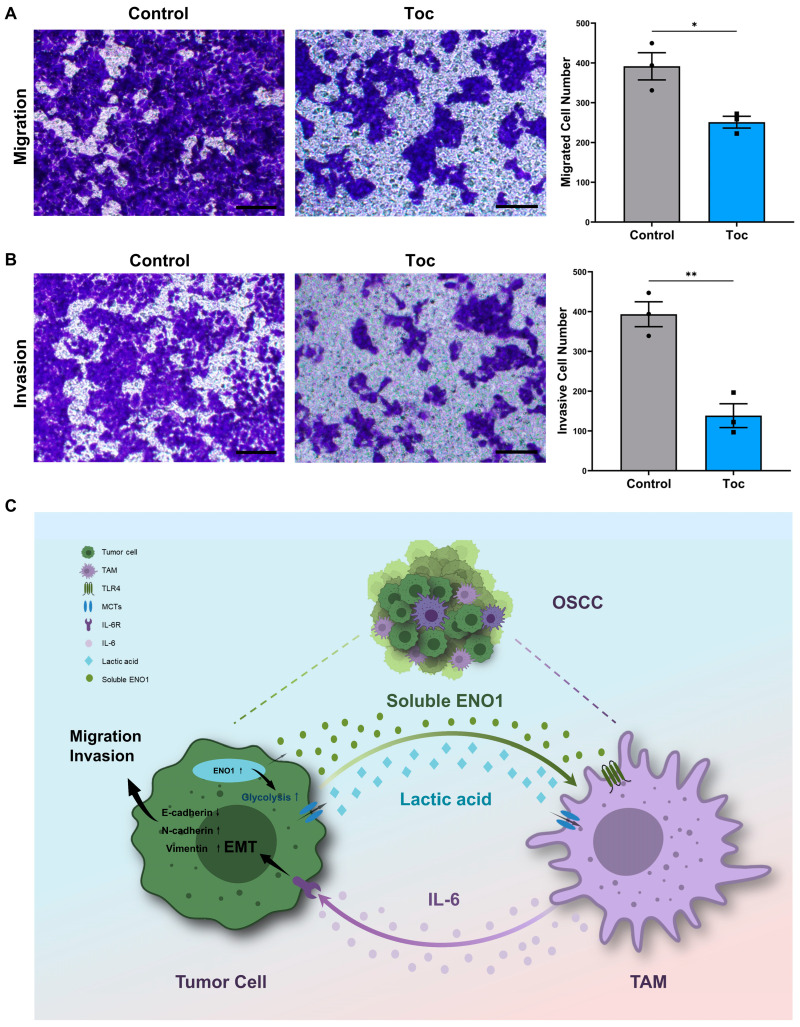
IL-6 promotes the migration and invasion of tumor cells. (**A**) Transwell assay for migration of CAL27 cells (upper chamber) pretreated without or with IL-6R antagonist tocilizumab (Toc, 5 μg/mL) and cocultured with macrophages (lower chamber) for 24 h. Left panel, representative pictures; right panel, bar charts indicating the cell numbers of migrated cells per area. Magnification 200×. Scale bar: 100 μm. (**B**) Transwell assay for invasion of CAL27 cells (upper chamber) pretreated without or with the IL-6R antagonist tocilizumab (Toc, 5 μg/mL) and cocultured with macrophages (lower chamber) for 48 h. Left panel, representative pictures; right panel, bar charts indicating the cell numbers of invasive cells per area. Magnification 200×. Scale bar: 100 μm. (**C**) Graphical abstract. ENO1 promotes tumor cell migration, invasion and EMT by orchestrating macrophage-derived IL-6 via secretion of lactic acid and extracellular ENO1 in OSCC, thus forming a positive feedback loop to promote OSCC progression. Different symbols (circle/square/triangle) were used to represent the data points of independent biological repeated experiments. All data are displayed as mean ± SEM; n = 3; * *p* < 0.05 and ** *p* < 0.01.

## Data Availability

Not applicable.

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
