# Peer review of "ENO1 Promotes OSCC Migration and Invasion by Orchestrating IL-6 Secretion from Macrophages via a Positive Feedback Loop"

_ijms, 2023, doi:10.3390/ijms24010737_

Round 1

Reviewer 1 Report

In this manuscript, the authors aimed to explore the regulatory mechanism of tumor-cell derived alpha-enolase (ENO1) in the interaction between tumor cells and TAMs during OSCC invasion and metastasis. Their results indicated that  ENO1 can promote migration and invasion of tumor cells by facilitating the epithelial-mesenchymal transition (EMT) through TAMs. ENO1 can increase IL-6 secretion of TAMs by tumor cell-derived lactic acid and paracrine ENO1/toll-like receptor signaling pathway. Overall, the study was well-designed and the results were clearly presented. But there are some points should be addressed before it can be accepted for publication.

1.           In Figure 2, the cells in the images of transwell assay were too small to be counted. I think the magnification should be 200X instead 40X. Please replace the images with higher magnification.

2.           In Figure 6, the colocalization of ENO1 and TLR4 was difficult to see. The authors should showed some images with higher magnification. And whether the addition of rhENO1 can increase the colocalization should be checked.

3.           The catalog number, the dilution of the antibodies used in the experiment of IF should be indicated too.

4.           There are some typos and grammar errors in the text, and the gene names should be showed correctly. Please check and revise the manuscript carefully.

Author Response

The authors thank the reviewers for spending considerable time and effort on examining this manuscript and greatly appreciate the constructive comments and suggestions made by the reviewers. We have included our point-by-point response to each comment and revisions to the manuscript have been marked up using the “Track Changes” function in MS Word.

Each comment brought up by the reviewers is in italics and our responses are in red.

In this manuscript, the authors aimed to explore the regulatory mechanism of tumor-cell derived alpha-enolase (ENO1) in the interaction between tumor cells and TAMs during OSCC invasion and metastasis. Their results indicated that  ENO1 can promote migration and invasion of tumor cells by facilitating the epithelial-mesenchymal transition (EMT) through TAMs. ENO1 can increase IL-6 secretion of TAMs by tumor cell-derived lactic acid and paracrine ENO1/toll-like receptor signaling pathway. Overall, the study was well-designed and the results were clearly presented. But there are some points should be addressed before it can be accepted for publication.

Point 1: In Figure 2, the cells in the images of transwell assay were too small to be counted. I think the magnification should be 200X instead 40X. Please replace the images with higher magnification.

Response 1:  The original images have been replaced by that with higher magnification (200X) in the revised manuscript.

Point 2: In Figure 6, the colocalization of ENO1 and TLR4 was difficult to see. The authors should showed some images with higher magnification. And whether the addition of rhENO1 can increase the colocalization should be checked.

Response 2: Representative images with higher magnification (400X) have been shown in the revised manuscript. And the experiment has been repeated to verify the addition of rhENO1 can increase the colocalization of ENO1/TLR4 in macrophages.

Point 3: The catalog number, the dilution of the antibodies used in the experiment of IF should be indicated too.

Response 3: The primary antibodies were used in this experiment: anit-ENO1 (ab155102, 1:500) and anti-TLR4 (ab22048, 1:100) from abcam. The secondary antibodies were used in this experiment: Goat anti-Rabbit IgG (H+L) conjugated with Alexa Fluor 594 (R37117, 2 drops diluted in 1ml PBS) from Thermo Fisher Scientific and goat anti-mouse IgG (H+L) conjugated with Alexa Fluor 488 (ab150113, 1:200) from abcam.

Point 4: There are some typos and grammar errors in the text, and the gene names should be showed correctly. Please check and revise the manuscript carefully.

Response 4: We have checked and revised the manuscript carefully. Revisions to the manuscript have been marked up using the “Track Changes” function in MS Word. Meanwhile, the gene names were corrected in table S2 and figure 4.

Reviewer 2 Report

Comments and Suggestions for Authors:

Lin et al. proposed a potential interesting study regarding the crosstalk between OSCC cells and macrophages, revealing a dual mechanism through the positive feedback loop of ENO1-TLR4-IL6 to promote the migration and invasion of OSCC cancer cells. Quite a few questions need to be addressed clearly and properly, and major revision needs to be done before the manuscript could be considered for publication in IJMS.

Reviewer’s comments:

1.     The title of the manuscript is not very clear with the study presented here.

First, RAW264.7 is a mouse macrophage cell line, and it needs to be treated with conditional medium (CM) from cancer cells to be active and induced to become TAMs, so that the authors should not consider this cell line as TAMs automatically without pre-treated with conditional medium (CM) from CAL27 cells or other OSCC cells. It is not properly to use the terms of TAMs throughout the study, including the title. The authors need to change the term of TAM to macrophages or explain whether RAW264.7 cells have been treated with CM from any cancer cells to be able to use TAMs in this study?

Second, there is no metastasis study in this manuscript, although the EMT experiments were presented in Result 2.3, the concept is not the same, so the terms of metastasis that appeared in the title were improperly.

Finally, the authors were trying to tell us about the crosstalk between OSCC cells and TAMs. Provided that both lactic acid, which was produced by ENO1 in OSCC, and the extracellular secreted protein of ENO1 derived from OSCC can elevate IL6 production by TAM, which promote OSCC cell migration and invasion in a positive feedback loop. But the current title does not reflect the study content and needs to be modified properly.

2.     The authors need to specify the different between the CAL27 and HaCaT cell line either in the Introduction or in the first result section.

3.     How many siRNAs were used in this study? In general, at least 3 different siRNA oligos should be used to avoid the off-target effects of siRNA.

4.     How to explain why mRNA levels change but not protein levels? (Line72-73)

5.     In lines 103-104, the authors said: 'The wound healing assay shows that TAMs-CM from ENO1 siRNA transfected TCM-induced macrophages significantly decreased the horizonal migration capacity of CAL27 cells’, what is that mean? Please describe in an accurate sentence.

6.     What is si-NC Figure 1F and in line149? Is it scramble RNA? Please use appropriate description.

7.     These experiments were to examine the effect of the CM from ENO1 knockdown Cal27 cells to macrophages, the demonstrated effect of IL6 could be the target of ENO1 in macrophages not in OSCC. (Line 166)

8.     How to explain the change in IL10 level in Figure 4?

9.     TLR4 should be in cell membranes but is shown in nuclear staining in Figure 6?

10.  In Section 2.7 the authors used tocilizumab as IL6 antagonist to treat OSCC, so the authors should not say that IL6 was TAM-derived? (Line 239). This section can only demonstrate that the effect of IL6 on OSCC is nothing about TAMs.

11.  The experiment in Figure 7A (Line 245-246) does not make sense since the authors have suggested that IL-6 secreted by macrophages can change OSCC cell behaviour and what do they expect when IL6 antagonist is added and then cocultured with macrophages again? Does it like the authors added an IL6 antagonist and had macrophages secret IL6 the same time and what did the author expect to see in this controversial experimental design?

12.  The English language needs to be revised to present it more clearly throughout the manuscript.

Author Response

The authors thank the reviewers for spending considerable time and effort on examining this manuscript and greatly appreciate the constructive comments and suggestions made by the reviewers. We have included our point-by-point response to each comment and revisions to the manuscript have been marked up using the “Track Changes” function in MS Word.

Each comment brought up by the reviewers is in italics and our responses are in red.

Point 1: The title of the manuscript is not very clear with the study presented here.

First, RAW264.7 is a mouse macrophage cell line, and it needs to be treated with conditional medium (CM) from cancer cells to be active and induced to become TAMs, so that the authors should not consider this cell line as TAMs automatically without pre-treated with conditional medium (CM) from CAL27 cells or other OSCC cells. It is not properly to use the terms of TAMs throughout the study, including the title. The authors need to change the term of TAM to macrophages or explain whether RAW264.7 cells have been treated with CM from any cancer cells to be able to use TAMs in this study?

Second, there is no metastasis study in this manuscript, although the EMT experiments were presented in Result 2.3, the concept is not the same, so the terms of metastasis that appeared in the title were improperly.

Finally, the authors were trying to tell us about the crosstalk between OSCC cells and TAMs. Provided that both lactic acid, which was produced by ENO1 in OSCC, and the extracellular secreted protein of ENO1 derived from OSCC can elevate IL6 production by TAM, which promote OSCC cell migration and invasion in a positive feedback loop. But the current title does not reflect the study content and needs to be modified properly.

Response 1: According to the reviewer’s suggestions, the title has been changed to “ENO1 Promotes Tumor Cell Migration and  Invasion by Orchestrating Macrophage-derived IL-6 via Secretion of Lactic Acid and Extracellular ENO1 in OSCC”. Meanwhile, the term “invasion and metastasis” has been changed to “cell migration and invasion” in the main text. And the term “TAM” has been changed to “macrophage” throughout the study, including the title.

Point 2: The authors need to specify the different between the CAL27 and HaCaT cell line either in the Introduction or in the first result section.

Response 2: The differences between the CAL27 and HaCaT cell line have been added in the first result section as follows. CAL27 is a human tongue squamous cell carcinoma cell line. HaCaT is an immortalized normal epithelial cell line that is often used as a normal control when compared with CAL27 cells. (page 2, line 28)

Point 3: How many siRNAs were used in this study? In general, at least 3 different siRNA oligos should be used to avoid the off-target effects of siRNA.

Response 3: Three siRNA targeting ENO1 were used in this study. The figures showed the results of the siRNA-1 with the highest transfection efficiency. Sequences of the siRNAs are listed as follows.

Point 4: How to explain why mRNA levels change but not protein levels?

Response 4: Figure 1 showed mRNA levels of ENO1 changed but not protein levels detected by western blot. Factors that explain why mRNA levels changed but not protein levels detected by western blot are listed as follows. First, ENO1 can be located in the nucleus, cytoplasm and cell membrane, as well as the extracellular environment. Furthermore, total protein levels of ENO1 may not be changed, but secretion levels were regulated by TAMs-CM. On the other hand, ENO1 was wrapped in exosomes [1], and differences in regular protein lysis could not be detected by Western blot.

Point 5: In lines 103-104, the authors said: 'The wound healing assay shows that TAMs-CM from ENO1 siRNA transfected TCM-induced macrophages significantly decreased the horizonal migration capacity of CAL27 cells’, what is that mean? Please describe in an accurate sentence.

Response 5: This part has been re-written as follows. “Then, RAW264.7 cells were stimulated with tumor-conditioned medium (TCM) of transfected CAL27 cells. Next, CM from macrophages in different groups (scrambled siRNA and ENO1 siRNA) was collected and used to incubate CAL27 cells. The wound-healing assay showed that the migration capacity of treated CAL27 cells was significantly decreased in the incubation with CM of ENO1 siRNA-treated group.” (page 4, line 5-8)

Point 6: What is si-NC Figure 1F and in line149? Is it scramble RNA? Please use appropriate description.

Response 6: si-NC refers to siRNA of negative control (NC). It is scramble siRNA, which is used as a non-target siRNA treatment group. In the revised manuscript, “si-NC” in Figure 1F and other Figures was changed to “scramble”. “si-NC” in the text has also been corrected to “scrambled siRNA”. (page  16, line 13)

Point 7: These experiments were to examine the effect of the CM from ENO1 knockdown Cal27 cells to macrophages, the demonstrated effect of IL6 could be the target of ENO1 in macrophages not in OSCC. (Line 166)

Response 7: We are sorry for the lack of clarity in the text. This sentence has been revised as follows. These results indicate that macrophage-derived IL-6 is a potential downstream target of ENO1. (page  8, line 14)

Point 8: How to explain the change in IL10 level in Figure 4?

Response 8: Figure 4 showed the mRNA level change of IL10 in the early stage of CM incubation (6h and 12h), and the mRNA level change may not be synchronized with the protein level changes, which could not represent the overall change trend of IL10 in the condition of CM incubation. Generally, the results showed that ENO1 inhibited IL-10 mRNA levels. However, since IL-10 is not a key cytokine in this study, the mechanism of ENO1 regulating IL-10 has not been revealed yet. This will be the content of our subsequent study.

The change in IL10 level in figure 4 could be explained as follows.

In this part of the experiment, macrophages were stimulated with tumor- conditioned medium (TCM) supplemented without or with rhENO1. Firstly, TCM-activated macrophages are a mixed group of M1 and M2-like subsets [1]. There are some factors to promote M1 or M2-like phenotype in TCM in vitro. Overall, the cytokine expression profiles of TAMs are spatially and temporally diverse. Secondly, ENO1 promotes M1-like polarization in the early stage. The overall effect is to promote tumor, as M1-like TAMs could cascade a mesenchymal/stem-like phenotype of OSCC via the IL6/Stat3/THBS1 feedback loop [2] It seems not contradictory to ENO1. Thirdly, even though lactic acid promotes IL10 expression, there may be a time gap between lactic acid production and the effect of rhENO1 on IL-10 in macrophages under the incubation of tumor-conditioned medium. Fourthly, the expression of different cytokines has a time difference, and in the early stage, the IL-10 we examined is the inhibitory level. It's not synchronized with IL-6. In the model of rheumatoid arthritis (RA), ENO1 induces early production of pro-inflammatory cytokines and chemokines but delayed production of IL-10 to activate the innate immune system[3].

Point 9: TLR4 should be in cell membranes but is shown in nuclear staining in Figure 6?

Response 9: Nonspecific staining of nuclear may result from the expiration of TLR4 antibodies. The IF experiment has been repeated and the representative images with higher magnification have been changed in the revised manuscript.

Point 10: In Section 2.7 the authors used tocilizumab as IL6 antagonist to treat OSCC, so the authors should not say that IL6 was TAM-derived? (Line 239). This section can only demonstrate that the effect of IL6 on OSCC is nothing about TAMs.

Response 10: This section has been revised according to the reviewer’s comments. The description” TAM-derived” has been deleted in this section.

Point 11: The experiment in Figure 7A (Line 245-246) does not make sense since the authors have suggested that IL-6 secreted by macrophages can change OSCC cell behaviors and what do they expect when IL6 antagonist is added and then cocultured with macrophages again? Does it like the authors added an IL6 antagonist and had macrophages secret IL6 the same time and what did the author expect to see in this controversial experimental design?

Response 11:  The results section of figure 7 was revised more clearly.

In this part, we used an IL-6 receptor antagonist (tocilizumab) to pre-treat OSCC cells. Macrophages seeded on the low chamber were incubated with tumor-conditioned medium in advance and then cocultured with the OSCC cells mentioned above. CAL27-CM-activated macrophages have higher IL-6 secretion levels. As a previous study reported, breast cancer-conditioned medium activates macrophages and enhances IL-6 expression [4]. We try to figure out whether the inhibition of IL-6 receptors could attenuate the interaction of macrophage and OSCC cells to some extent. Lastly, Figure 7 A could be attached to a supplemental figure if necessary.

Point 12: The English language needs to be revised to present it more clearly throughout the manuscript.

Response 12: Revisions to the manuscript have been marked up using the “Track Changes” function in MS Word to present it more clearly.

  1. Jiang, M.; Qi, Y.; Huang, W.; Lin, Y.; Li, B. Curcumin Reprograms TAMs from a Protumor Phenotype towards an Antitumor Phenotype via Inhibiting MAO-A/STAT6 Pathway. Cells 2022, 11, doi:10.3390/cells11213473.
  2. You, Y.; Tian, Z.; Du, Z.; Wu, K.; Xu, G.; Dai, M.; Wang, Y.; Xiao, M. M1-like tumor-associated macrophages cascade a mesenchymal/stem-like phenotype of oral squamous cell carcinoma via the IL6/Stat3/THBS1 feedback loop. J Exp Clin Cancer Res 2022, 41, 10, doi:10.1186/s13046-021-02222-z.
  3. Guillou, C.; Freret, M.; Fondard, E.; Derambure, C.; Avenel, G.; Golinski, M.L.; Verdet, M.; Boyer, O.; Caillot, F.; Musette, P.; et al. Soluble alpha-enolase activates monocytes by CD14-dependent TLR4 signalling pathway and exhibits a dual function. Sci Rep 2016, 6, 23796, doi:10.1038/srep23796.
  4. Radharani, N.N.V.; Yadav, A.S.; Nimma, R.; Kumar, T.V.S.; Bulbule, A.; Chanukuppa, V.; Kumar, D.; Patnaik, S.; Rapole, S.; Kundu, G.C. Tumor-associated macrophage derived IL-6 enriches cancer stem cell population and promotes breast tumor progression via Stat-3 pathway. Cancer Cell Int 2022, 22, 122, doi:10.1186/s12935-022-02527-9.

Round 2

Reviewer 1 Report

All my concerns were well addressed by the authors and the manuscript was improved. I think it is fine for publication now.

Author Response

Thank you for your constructive comments and suggestions. We are grateful for your support of our work.

Reviewer 2 Report

1. Potential Title: ENO1 promotes OSCC migration and invasion by orchestrating IL-6 secretion from macrophages via a positive feedback loop

2. Please incorporate the explanation of IL-10 expression (point 8) in the discussion part.
